# Genomic Epidemiology of Salmonid Alphavirus in Norwegian Aquaculture Reveals Recent Subtype-2 Transmission Dynamics and Novel Subtype-3 Lineages

**DOI:** 10.3390/v13122549

**Published:** 2021-12-20

**Authors:** Daniel J. Macqueen, Oliver Eve, Manu Kumar Gundappa, Rose Ruiz Daniels, Michael D. Gallagher, Svein Alexandersen, Marius Karlsen

**Affiliations:** 1The Roslin Institute and Royal (Dick) School of Veterinary Studies, The University of Edinburgh, Edinburgh EH25 9RG, UK; Oliver.Eve@ed.ac.uk (O.E.); manu.gundappa@roslin.ed.ac.uk (M.K.G.); Rose.Daniels@roslin.ed.ac.uk (R.R.D.); 2Bionano Genomics, 9540 Towne Centre Dr #100, San Diego, CA 92121, USA; mgallagher@bionanogenomics.com; 3Pharmaq AS, 0275 Oslo, Norway; svein.alexandersen@zoetis.com (S.A.); Marius.Karlsen@zoetis.com (M.K.)

**Keywords:** genomic epidemiology, genomic surveillance, viral genomics, aquaculture, salmonid alphavirus, pancreas disease

## Abstract

Viral disease poses a major barrier to sustainable aquaculture, with outbreaks causing large economic losses and growing concerns for fish welfare. Genomic epidemiology can support disease control by providing rapid inferences on viral evolution and disease transmission. In this study, genomic epidemiology was used to investigate salmonid alphavirus (SAV), the causative agent of pancreas disease (PD) in Atlantic salmon. Our aim was to reconstruct SAV subtype-2 (SAV2) diversity and transmission dynamics in recent Norwegian aquaculture, including the origin of SAV2 in regions where this subtype is not tolerated under current legislation. Using nanopore sequencing, we captured ~90% of the SAV2 genome for n = 68 field isolates from 10 aquaculture production regions sampled between 2018 and 2020. Using time-calibrated phylogenetics, we infer that, following its introduction to Norway around 2010, SAV2 split into two clades (SAV2a and 2b) around 2013. While co-present at the same sites near the boundary of Møre og Romsdal and Trøndelag, SAV2a and 2b were generally detected in non-overlapping locations at more Southern and Northern latitudes, respectively. We provide evidence for recent SAV2 transmission over large distances, revealing a strong connection between Møre og Romsdal and SAV2 detected in 2019/20 in Rogaland. We also demonstrate separate introductions of SAV2a and 2b outside the SAV2 zone in Sognefjorden (Vestland), connected to samples from Møre og Romsdal and Trøndelag, respectively, and a likely 100 km Northward transmission of SAV2b within Trøndelag. Finally, we recovered genomes of SAV2a and SAV3 co-infecting single fish in Rogaland, involving novel SAV3 lineages that diverged from previously characterized strains >25 years ago. Overall, this study demonstrates useful applications of genomic epidemiology for tracking viral disease spread in aquaculture.

## 1. Introduction

Modern genomic tools have revolutionized how viral pathogens are studied and monitored in human and animal populations [1]. Viruses accumulate mutations rapidly and sequencing can be used to trace how genetic changes are distributed among infected individuals; when sampled through time and space, such data can be applied to make epidemiological inferences on disease transmission. Fuelled by recent advances in second and third-generation sequencing, including portable devices such as the MinION platform (Oxford Nanopore Technologies, ONT), genomic sequencing and surveillance programs have gained prominence as tools for monitoring human viral diseases [2,3]. Such efforts are best illustrated by global-scale initiatives to sequence hundreds of thousands of SARS-CoV-2 genomes in human populations, which have been instrumental in characterizing genetic diversity and COVID-19 transmission routes [4,5].

The global aquaculture sector is expanding every year to meet the growing nutritional demands of the rapidly growing human population [6]. It is well recognized that disease outbreaks are one of the main barriers to the sustainable expansion of aquaculture [7]. Viral pathogens pose a particular challenge due to limited effective treatment options, coupled with the fact that outbreaks are highly damaging financially and in terms of animal welfare. A range of approaches are available to prevent and control viral disease in aquaculture [8], including selective breeding [9] and vaccination for finfish [10]. While it is recognized that genomic epidemiology can deliver much to support disease control efforts in aquaculture [11] and other sectors [12], large-scale genomic sequencing to monitor aquaculture pathogens is not a widespread practice, even for high-value species. Nonetheless, several studies highlight the value of genomic epidemiology for linking viral genetic diversity to disease outbreaks and identifying viral transmission scenarios [13,14,15,16,17].

Atlantic salmon (*Salmo salar*) is globally the highest value aquaculture species [9,18], with the Norwegian sector producing 1.3 million tonnes across almost 1000 sites in 2019, valued at 68 billion NOK (5.6 billion GBP) [19]. Among several viral pathogens infecting farmed salmonids [20], SAV was first recognized as the agent of Atlantic salmon PD in the 1990s [21,22,23,24] and shortly after sleeping disease (SD) in rainbow trout (*Oncorhynchus mykiss*) [25]. PD and SD cause related pathologies characterized by damage to several organs, including pancreatic necrosis and striated muscle inflammation, associated with slow growth, mortality, and in surviving fish, poor flesh quality [26]. SAV is a single-stranded positive-sense RNA virus with a genome size of ~12,000 bp, belonging to the genus *Alphavirus* within *Togiviridae* [27]. It comprises seven defined phylogenetic groups (hereafter: subtypes; SAV1-SAV7) [28,29]. Among these, SAV1-6 infect Atlantic salmon, and were introduced independently to aquaculture from wild reservoirs, followed by subsequent spread within the industry [30]. SAV is endemic in European aquaculture, with the various subtypes showing partly overlapping distribution in different countries [27]. SAV has also been detected in wild reservoir species, including flatfishes and ballan wrasse [15,29,31].

PD has been present in Norway since the 1980s and is thought to have been exclusively caused by SAV3 until SAV2 was detected during an outbreak in 2010 [32,33]. SAV2 was likely initially transmitted to mid-Norway from Scotland, where SAV2 is a common PD-causing subtype [34]. This may have occurred via industry well boat activity, though transmission from wild reservoir species cannot yet be ruled out [34]. SAV2 and SAV3 are defined as separate endemics in Norway, tolerated in largely non-overlapping regions within a wider endemic PD zone, stretching from the southern end of Norway to Flatanger in Trøndelag [35] (Figure 1). Detection of PD outside the endemic zone triggers immediate disease control actions [35]. Within the endemic zone, SAV3 is predominantly detected in Hordaland and Rogaland, while SAV2 dominates in Møre og Romsdal and Trøndelag [36]. With the aim of restricting the spread of SAV2 and SAV3, the endemic PD zone is currently split at Hustadvika (Møre og Romsdal), with SAV2 and SAV3 tolerated above and below this boundary, respectively [35]. SAV2 has recently been detected outside the endemic PD zone and within the SAV3 endemic zone [36], but the origin of these outbreaks is unestablished.

The primary aim of this study was to reveal SAV2 genetic diversity and PD transmission dynamics in recent Norwegian aquaculture using a genomic epidemiology approach. We used nanopore sequencing to generate SAV2 genomic sequences from infected fish sampled between 2018 and 2020 from 27 sampling sites spanning 10 major production areas (Figure 1). Our previous work has shown that nanopore sequencing of SAV and other fish viruses offers a rapid and cost-effective approach with extremely high consensus accuracy, which can be used to identify complex infections involving different viral strains [14,37]. Using extensive new nanopore data alongside existing SAV genome sequence data, we reconstruct the phylogenetic diversity of SAV2 circulating in recent Norwegian aquaculture, revealing plausible geographical origins for recent SAV2 outbreaks outside the SAV2 endemic zone, and identify novel ancestral lineages of SAV3 in fish co-infected with both SAV2 and SAV3.

## 2. Material and Methods

### 2.1. Overview of SAV Nanopore Sequencing Approach

We adapted our published strategy [14,37] to generate SAV genomic sequence data used in this study. As summarized in Figure 2, this involves PCR amplification of the SAV genome in six products, using primers designed to cross-amplify all known SAV subtypes (primer sequences in Table 1), followed by multiplex nanopore sequencing on the ONT MinION platform (Oxford Nanopore Technologies, Oxford, UK), with samples identified by a unique barcode. The six primer pairs generated overlapping PCR products in all cases except the 3’ and 5’ respective boundaries of amplicon 4 and 5 (Figure 2), as we were unable to generate amplicons bridging ORF1 (encoding the non-structural proteins) and ORF2 (encoding the structural proteins) despite extensive trials using multiple primer pairs.

### 2.2. Samples, Reverse Transcription and PCR 

Samples were sent by Pharmaq AS to the Roslin Institute on dry ice in July 2020. All samples were total RNA extracted from heart tissue from individual fish infected with SAV according to a commercial qPCR diagnostic test. In advance of shipping, RNA extractions were done by Pharmaq Analytiq or Patogen using standard commercial approaches applied for routine diagnostics. Full metadata for samples are provided in Appendix A, including the isolate name, production region, sampling date, estimated heart viral load (qPCR Cq value), ONT barcode (Section 2.4), and sequence mapping statistics (Section 2.4).

Total RNA samples were transferred into a 96 well PCR plate, inclusive of 94 samples and 2 negative controls (nuclease-free water) (Figure 2). We quantified the concentration of each sample using a Qubit 3 fluorimeter with an RNA high sensitivity assay kit (Thermo Fisher Scientific, Waltham, MA, USA) and confirmed total RNA integrity via agarose gel electrophoresis. First strand cDNA synthesis was performed in a plate-set up with standardized total RNA input across samples (0.4 µg) using Protoscript II Reverse Transcriptase (New England Biolabs, Hitchin, UK) in 20 µL reactions following the manufacturer’s protocol, including a mix of random hexamer (1.25 µM concentration) and anchored-dT (dT_23_VN) primers (1.5 µM concentration) and 400 U enzyme, with the following conditions: 25 °C for 5 min, 42 °C for 1 h, and 80 °C for 5 min. The 1:1 first strand cDNA was diluted 20× with nuclease-free water to make a working stock for PCR.

PCRs were done in six separate plates (one per amplicon; Figure 2) using Phusion Hot Start II High-Fidelity PCR Master Mix (Thermo Fisher Scientific) according to the manufacturer’s protocol. Each 20 µL reaction contained 10 µL of 2× Master Mix, 0.25 µM each of forward and reverse primer (Table 1), and 2 µL of working stock cDNA. The PCR cycling conditions were: 1 cycle of 98 °C for 1 min; 35 cycles of 98 °C for 15 s, 56 °C for 1 min, 72 °C for 2 min 15 s; and 7 min at 72 °C. Prior to performing plate PCRs, a pool of all 94 cDNA samples was generated and used to confirm a PCR product of the expected size was generated using agarose gel electrophoresis for each primer set, with the above stated PCR conditions. PCR reactions for 12 (out of the 94) samples were run for each amplicon on a Tapestation 4200 using D5000 ScreenTapes (Agilent Technologies, Santa Clara, CA, USA) to confirm amplification in the expected size range. The concentration of the same 12 PCR samples for each target amplicon was quantified on a Qubit 3 fluorimeter using a high sensitivity DNA kit (Thermo Fisher Scientific). The average yield of DNA in the 12 samples was used to approximately normalize the input of DNA across amplicons when generating a final pool containing all six amplicons per sample. This aimed to avoid over-dominant sequencing of regions in the SAV genome that amplified more efficiently during PCR. For samples (including both negative controls), where there was insufficient DNA for an amplicon to meet the targeted normalized level, the whole PCR reaction was included in the pool. Finally, the pooled PCR reactions for the 94 samples plus two negative controls were individually cleaned using AMPure XP beads (Beckman Coulter, High Wycombe, UK) with a 0.6 bead-to-PCR reaction ratio (selecting for amplicons 1–3 kb pair in size). The cleaned samples were quantified with a Qubit 3 fluorimeter using a high sensitivity DNA kit (Thermo Fisher Scientific). Equimolar pooling was performed across all samples to obtain 1.5 µg of DNA in total (~16 ng per pooled sample), which was the basis for ONT sequencing (next section).

### 2.3. ONT Library Preparation, MinION Sequencing, Basecalling, and Trimming

0.2 µg of the pooled amplicons covering all 94 samples was the input to the ONT library preparation, using the 1D Ligation Sequencing Kit (SQK-LSK109) and the Native Barcoding Expansion 96 kit (EXP-NBD196). The NEBNext Ultra II End-Repair/dA-tailing Module (New England Biolabs, E7546) was first used to repair and dA-tail the ends of the DNA. Next, ONT’s native barcodes 1–96 were blunt-end ligated onto each sample using Blunt/TA Ligase Master Mix (New England Biolabs). 5 µL of each barcoded sample was pooled and taken forward to AMPure XP (Beckman Coulter) bead purification in a 1:2.5 ratio of beads to library. The beads were washed twice using 700 µL of short fragment buffer (SFB-ONT) before a final wash in 100 µL of freshly-prepared 80% ethanol. The barcoded, pooled library was eluted in 35 µL of nuclease-free water and 1 µL was quantified using a Qubit3 fluorometer using a high sensitivity DNA kit. 0.125 µg of DNA was carried forward for adapter ligation. AMII sequencing adapters (ONT) were ligated to the sequencing library using the NEBNext Quick Ligation Module (New England Biolabs, E6056) before purification in a 1:2.5 ratio of AMPure beads. The beads were washed twice with 125 µL short fragment buffer and then eluted in 15 µL of ONT elution buffer. 1 µL of the adapter-ligated library was quantified with a Qubit 3 fluorimeter using a high sensitivity DNA kit, before 100 ng of the pooled library was sequenced using flowcell chemistry R9.4.1 on a MinION Mk1B for 24 h without real-time basecalling.

Basecalling of raw fast5 files was carried out on a Unix GPU system with Guppy v4.2.2 using the high-accuracy basecalling model. All reads with PHRED score <7 were removed using NanoFilt [38] before demultiplexing was done using Guppy v4.2.2. ONT barcodes and adapters were trimmed using Porechop v0.2.4 [39]. The resulting fastq files were taken forward to mapping (Section 2.4).

### 2.4. Mapping, Consensus Sequence Generation, and Filtering

The ONT reads for each barcoded sample were mapped independently to full-length references genome sequences (NCBI accessions provided in parentheses) for the seven SAV subtypes: SAV1 (JX163854), SAV2 (MH708652 and AJ316246), SAV3 (AY604238), SAV4 (MH708651), SAV5 (MH708653), (SAV6: MH238448) and SAV7 (MT882199). Mapping was done with Minimap2 [40] using the parameter *-ax map-ont*, with other parameters set to default. SAMtools [41] was used to retrieve sequencing coverage and mean depth per SAV genome. Initially, the sorted bam files were parsed to retrieve genome-wide mapping coverage and mean depth values for all eight SAV genomes, revealing high-coverage mapping of most samples to SAV2 (MH708652) and for a few samples to SAV3 (AY604238) (see Section 3.1, Results). Using the same strategy, we retrieved per amplicon coverage and mapping depth for SAV2 (MH708652) and SAV3 (AY604238) across all samples (mapping depth shown in Appendix A). Consensus genomic sequences were then generated separately for SAV2 and SAV3 per sample barcode using SAMtools and BCFtools [42]. Pileup information for each base pair across both the SAV2 and SAV3 genomes was generated using the SAMtools *mpileup* option. Resulting VCF files were converted to fastq sequences using the BCFtools *vcfutils.pl vcf2fq* script, setting the minimum sequencing depth to assign a base at 50×. Finally, we used amplicon mapping depth for SAV2 and SAV3 in negative control samples as an additional quality filter (Appendix A) to account for background PCR amplification. Specifically, we accepted twice the mapping depth of the negative control showing the highest mapping depth as the minimum sequencing depth per amplicon. We masked amplicons (i.e., converting consensus nucleotides to ‘*NNN*’) that did not pass this filter using BEDtools [43].

Plots for mean depth and coverage across genomes and amplicons were generated using the ggplot2 v3.3.2 R package [44]. The scripts used for mapping and consensus sequences generation are provided in the Appendix A. Consensus sequences for SAV2 or SAV3 across the 94 samples are provided in Appendix A. 68 samples where consensus SAV sequences were recovered for no less than 5 out of the 6 SAV amplicons were taken forward to phylogenetic analyses (next section).

### 2.5. Phylogenetic and Phylogeographic Reconstructions

Phylogenetic analyses were performed on two separate alignments. The first included SAV2 genomes, with the aim to reconstruct the genetic diversity and spatial-temporal transmission dynamics of SAV2. The second included SAV2 and SAV3 genomes, with the aim to elucidate the phylogenetic position of novel SAV3 genomes discovered during the study. For the SAV2 alignment, 68 SAV2 sequences generated by ONT sequencing (Appendix A) were aligned along with 13 additional SAV2 genomes (Appendix A) (four generated in this study by Sanger sequencing; Section 2.6) using Mafft v.7 [45,46] with default settings. For the SAV2 and SAV3 alignment, the same SAV2 sequences were aligned along with 42 SAV3 genomes, including 3 SAV3 genomes generated here by ONT sequencing (Appendix A), and 39 further SAV3 genomes (Appendix A), using the same approach. Sequence alignments for the SAV2 (81 sequences, 11,684 sites) and SAV2 + SAV3 analyses (123 sequences, 11,684 sites) are provided as Appendix A.

For the SAV2 analysis, we performed time-calibrated Bayesian phylogenetic reconstruction in BEAST v2.6.4 [47]. The best-fitting nucleotide substitution model was established during the run within the embedded bModelTest package [48]. The analysis applied tip-date time calibration (for samples generated outside this study where only sampling year was available, we took the middle day of the year as representative), an uncorrelated relaxed clock model [49], a tree prior assuming constant viral population size, and a phylogeographic model to reconstruct the most likely geographic location for each node [50]. A Markov chain Monte Carlo chain (MCMC) of 200 million generations was used, sampling every 20,000 generations. MCMC chain convergence was confirmed in Tracer v1.7.2, where effective sample sizes were >200 for all parameters [51]. A maximum clade credibility tree was generated in TreeAnnotator v2.6.4 after removing the first 1000 sampled trees as burn-in (leaving 9000 trees sampled after convergence). Using the same SAV2 alignment, we performed a maximum likelihood phylogenetic reconstruction using the IQ-TREE webserver [52] with automated nucleotide substitution model selection [53] and an ultrafast bootstrap algorithm [54] to generate branch support values. For the SAV2+SAV3 analyses, we performed a maximum likelihood phylogenetic reconstruction using the same IQ-TREE approach. The best-fitting IQ-TREE models were TIM2+F+I and TIM2+F+G4 for the SAV2 only and SAV2 + SAV3 datasets, respectively.

Google maps (https://www.google.com/maps/, accessed on 20 August 2021) was used to visualize the locations of sampling sites and regions, using latitude and longitude coordinates provided by Pharmaq AS (Oslo, Norway). All producer-specific information has been anonymized in this study, with information connecting viral sequences in space and time-restricted to farming regions.

### 2.6. Sanger Sequencing of Additional SAV2 Genomes 

Four additional SAV2 genomes (details in Appendix A) were sequenced by traditional Sanger sequencing. Viral RNA was first purified using a QIAamp Viral RNA Mini Kit using carrier RNA on the QIACube. Purified RNA was then reverse transcribed into cDNA using Superscript III First-strand synthesis supermix (Invitrogen, Renfrewshire, UK). The near full-length genome (Appendix A) was amplified in 19 overlapping amplicons by reverse transcriptase PCR using Hot Start Taq DNA Polymerase (Qiagen, Manchester, UK). Successful amplicons were verified by gel electrophoresis and submitted to LGC (Wesel, Germany) for Sanger sequencing. The sequenced amplicons were built into genome sequences manually based on overlapping regions.

## 3. Results 

### 3.1. Nanopore Sequencing of SAV from Norwegian Field Samples

Six PCR amplicons spanning the SAV genome were sequenced in SAV-infected heart tissue from 94 Atlantic salmon individuals (Figure 1 and Figure 2) on the ONT MinION platform. After 24 h of sequencing, 3,175,118 reads were generated and subsequently basecalled, of which 2,554,754 passed filtering (q-score >7) and showed N50 of 1897 bp. We demultiplexed 94.6% of the passed reads, allowing data for all samples to be recovered by a unique barcode.

Commercial qPCR diagnostic tests done prior to sample selection indicated that most PD infections would represent SAV2. However, our sequencing and bioinformatic strategy captures all SAV subtypes, and can identify co-infections where multiple subtypes are present in samples [14]. Mapping against all seven SAV subtypes revealed that the SAV sequences captured were indeed dominated by SAV2 (Figure 3A,B; Appendix A). To demonstrate the sensitivity of our approach, we mapped the data to two different SAV2 genomes, one from an Atlantic salmon sampled in 2007 (accession: MH708652) [37] and the other from a rainbow trout sampled in 1997 (accession: AJ316246). While these sequences share 97.9% nucleotide identity, almost all mapping was to MH708652 (Figure 3A), which is expected as Norwegian SAV2 genomes are more closely related to this sequence than AJ316246 [34]. Mean mapping depth to SAV2 was 1010.8× across the 94 samples (range: 12.5× to 4392.3×), while mean mapping coverage was 96.3% (range: 74.5% to 99.5%), with 86 samples showing >90% coverage (Appendix A). Mapping to other SAV subtypes was negligible except for three samples that showed high mapping depth and coverage to SAV3 (Figure 3A,B).

We interrogated finer mapping statistics for SAV2 and SAV3, separating out each amplicon per sample (Appendix A). In addition to using a global mapping depth of >50 as a cut-off for consensus base identification, we used the negative controls to set a further cut-off for acceptance of consensus sequences per amplicon (see Methods, Section 2.4). We recovered at least 5 SAV2 amplicons for 68 out of the 94 samples, with 6 amplicons successfully captured in 33 of these samples (Figure 3C). The mean number of bases recovered across these 68 samples, which were taken forward for further analysis, was 10,599 bp (SD: 799 bp) (Appendix A), equivalent to ~90% of the SAV2 genome. Amplicons 1, 2, 3, 4, 5, and 6 were recovered in 85, 90, 90, 64, 38, and 85 samples, respectively, with respective mean mapping depths of 1754×, 2655×, 1672×, 675×, 390×, and 236× (Appendix A). Thus, ORF1, encoding the non-structural proteins (Figure 2), amplified more efficiently in this study.

For the 3 samples (FR182796282, FR182796280, FR182796275; all from Rogaland; Appendix A) showing high mapping depth/coverage to SAV3 (Figure 3A), all six amplicons were successfully recovered, with a mean of 11,076 bp (SD: 975 bp) captured, equivalent to 94% of the SAV3 genome. Significant mapping was also observed against SAV2 (accession MH708652) for these 3 samples. For FR182796280 all 6 SAV2 amplicons were recovered, while for FR182796282 and FR182796275, 2 and 4 SAV2 amplicons were recovered, respectively. This result is consistent with co-infection of SAV2 and SAV3 in single fish heart samples and demonstrates the value of our strategy to capture both genomes in such situations.

### 3.2. SAV2 Genetic Diversity in Recent Norwegian Aquaculture

The 68 novel SAV2 sequences, along with 13 additional SAV2 genome sequences (Appendix A), were used in a time-calibrated Bayesian phylogenetic analysis (Figure 4; Appendix A). This analysis included 4 new SAV2 genomes generated by Sanger sequencing (Section 2.5; Appendix A), which were sampled in Norway between 2011 and 2017, providing further resolution to divergence time estimates. For comparison, we performed a non-time-calibrated maximum likelihood analysis, providing an alternative visualization of the data (Appendix A).

Consistent with a recent study [34], all Norwegian SAV2 sequences were monophyletic and descended from Scottish SAV2 sequences (Figure 4A; Appendix A). According to the Bayesian tree (Figure 4A), which employed a relaxed uncorrelated molecular clock model [49] assuming constant viral population size, the most recent common ancestor (MRCA) of Norwegian SAV2 existed during 2010 (95% HPD: 2009–2011). This estimate is consistent with the first time that SAV2 was detected in 2010 [33]. SAV2 then split off into two major lineages around 2013 (95% HPD: 2011–2016), hereafter named SAV2a and 2b (Figure 4A). The MRCA of the sampled SAV2a and 2b lineages existed in 2016 (respective 95% HPDs: 2013–2018 and 2013–2017) (Figure 4A).

Sampling locations for SAV2a and 2b are shown in Figure 4B. SAV2b was more frequently detected (48/68 samples) and extended more northerly than SAV2a, explaining all SAV2 detection in Northern Trøndelag and as southerly as Outer Sognefjord in Vestland. The northern and southern limit of SAV2a detection was at sites in Trøndelag (Hitra area) and Rogaland, respectively. SAV2a was also detected in middle Sognefjord in Vestland. We found that SAV2a and 2b were co-present in fish sampled from the same sites in Hitra (Trøndelag) and Smøla (Møre og Romsdal), either on the exact same date or within a very close sampling timeframe of a few days apart (Appendix A; also, compare Figure 5 and Figure 6).

### 3.3. Phylogeographical Reconstructions of SAV2

Our analysis incorporated a phylogeography model [50], allowing the most likely ancestral locations of nodes throughout the SAV2 tree to be reconstructed (Figure 5 and Figure 6). Most samples taken from geographically proximal regions clustered closely together in the tree, with exceptions indicating viral spread between different regions. As a caveat of our approach, it should be noted that any inferences made on viral transmission are based on the sampled SAV2 sequences, and it is possible that unsampled viruses exist from alternative geographical regions, which could alter our interpretations and provide alternative transmission scenarios.

**Figure 5 viruses-13-02549-f005:**
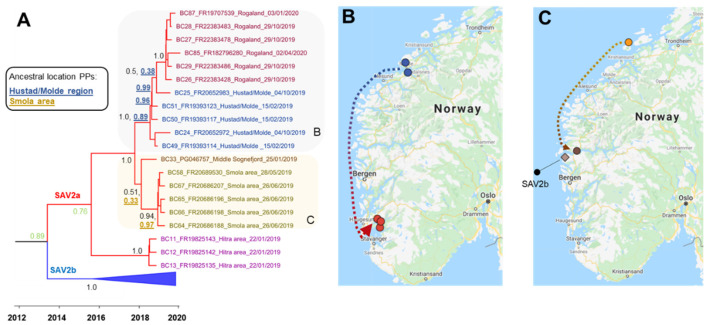
(**A**) Time-calibrated Bayesian phylogeny (Figure 4), focussing on the SAV2a clade, including results of the phylogeographic model. Tip names are colored according to the region of sample origin. PP values for key nodes are shown in black font. PP values for the ancestral location of samples are shown at relevant nodes, using font colors matching the given key for different aquaculture regions. (**B**) Inferred scenario of SAV2a spread from the Hustad/Molde area to Rogaland. (**C**) Inferred scenario of SAV2a spread from the Smøla area to Middle Sognefjord.

**Figure 6 viruses-13-02549-f006:**
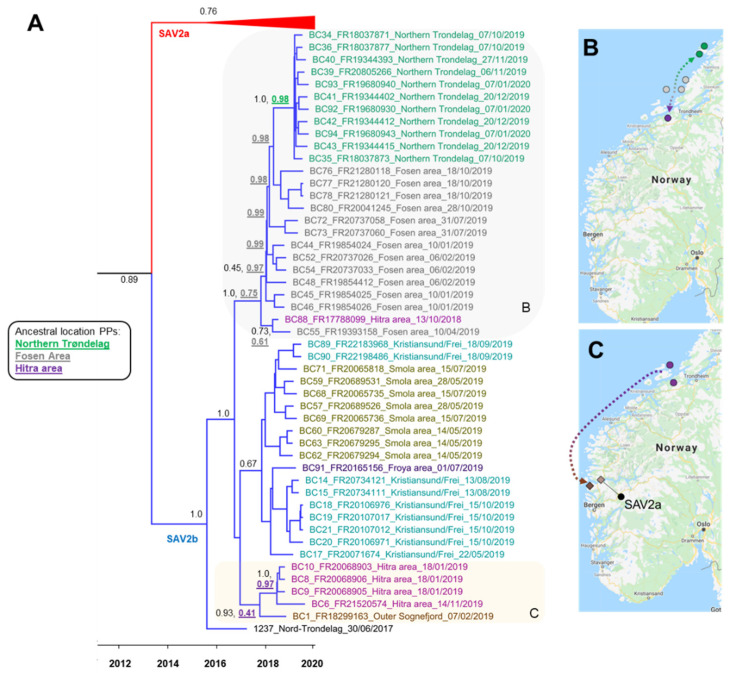
(**A**) Time-calibrated Bayesian phylogeny (Figure 4), focussing on the SAV2b clade, including results of the phylogeographic model. Tip names are colored according to the region of sample origin. PP values for key nodes are shown in black font. PP values for the ancestral location of samples are shown at relevant nodes, using font colors matching the given key for different aquaculture regions. (**B**) Inferred scenario of SAV2a spread from the Fosen area to Northern Trøndelag. (**C**) Inferred scenario of SAV2a spread from the Hitra area to Outer Sognefjord.

#### 3.3.1. Phylogeographical Reconstructions of SAV2a

Within the SAV2a clade, there is a strong connection between the Hustad/Molde region (Møre og Romsdal) and SAV2a detected in Rogaland in 2019/20 (Figure 5A,B). Specifically, all SAV2a sequences from Rogaland and the Hustad/Molde region comprise a sub-clade with maximal statistical support (posterior probability [PP]: 1.0). Within this clade, Rogaland sequences are monophyletic (PP: 1.0) and descended from samples in Hustad/Molde, which is strongly inferred as the ancestral location for the Rogaland + Hustad/Molde clade (Figure 5A). An SAV2a sequence from Hustad/Molde that branches as the immediate ancestor of Rogaland sequences (PP: 0.5) was sampled in late Oct 2019, 25 days before the earliest SAV2a samples taken in Rogaland (Figure 5A). 

Within the SAV2a clade, our analysis infers an introduction of this strain into Middle Sognefjord (Figure 5B). Specifically, an SAV2a sequence from Middle Sognefjord formed a sister group to a clade of Smøla (Møre og Romsdal) sequences (PP: 0.51) (Figure 5A). However, statistical support for Smøla being the source of SAV2a in Middle Sognefjord was weak according to the phylogeography model (PP: 0.33). As the clade containing Smøla and Middle Sognefjord sequences otherwise only contains sequences from Hustad/Molde (Møre og Romsdal) and Hitra (Trøndelag) (Figure 5A), it seems likely that Møre og Romsdal or Trøndelag was the source of virus detected in Middle Sognefjord.

#### 3.3.2. Phylogeographical Reconstructions of SAV2b

Within the SAV2b clade, a genome sampled in Northern Trøndelag in 2017 is the most basal branch (PP: 1.0) (Figure 6A). Two SAV2b subclades are supported (PP: 1.0), the first dominated by sequences from the Fosen area and Northern Trøndelag (Figure 6A). Within this clade, Northern Trøndelag sequences are monophyletic (PP: 1.0) and descended from sequences sampled from the Fosen area of Trøndelag, which is strongly recovered as the ancestral location of SAV2b introduced to Northern Trøndelag according to the phylogeography model (PP: 0.98) (Figure 6A,B). There is also evidence at the base of this subclade for transmission of SAV2b across different sites located in southern Trøndelag and Møre og Romsdal (Figure 6A,B).

The second SAV2b subclade is dominated by sequences from Trøndelag (Hitra, Fosen, and Frøya) and Møre og Romsdal (Kristiansund/Frei and Smøla) (Figure 6A). However, sitting at the base of this subclade, a sequence from Outer Sognefjord (Vestland) forms a sister relationship with a monophyletic group of Hitra sequences (PP: 0.93). Hitra was not strongly supported as the origin of SAV2b in Outer Sognefjord according to the phylogeography model (PP: 0.41). However, considering that the sister clade to the Hitra and Outer Sognefjord sequences solely includes SAV2b sequences from Trøndelag and Møre og Romsdal, it is likely that one of these counties was the source of SAV2b introduction to Outer Sognefjord in 2019, occurring independently from a separate introduction of SAV2a to Middle Sognefjord in a similar timeframe (Figure 5A,C).

### 3.4. Novel SAV3 Strains Detected in Fish Co-Infected with SAV2 and SAV3

A maximum likelihood phylogenetic tree was generated, including the three SAV3 genome sequences captured as co-infections with SAV2a in individual fish heart samples from Rogaland (Section 3.1). This was done in a framework covering all published SAV3 and SAV2 genomes, in addition to 72 SAV2 sequences generated in this study (Figure 7). The mid-point rooted tree recovered monophyletic SAV2 and SAV3 clades and the same overall branching of SAV2 sequences as the Bayesian tree (Figure 1; Appendix A). Within the SAV3 clade, the three novel genomes branched at the base of all previously published strains, which were sampled between 2002 and 2019. Furthermore, while FR182796282 and FR182796275 comprised a monophyletic group sister to all previously characterized SAV3 sequences, FR182796280 was, in turn, the sister sequence to all other SAV3 sequences, including FR182796282 and FR182796275 (Figure 7). A previous study revealed that the common ancestor of SAV3 genomes (excluding the sequences generated in this study) existed in 1994 [14]. Consequently, recently circulating SAV3 in Rogaland includes strains that shared a common ancestor with previously characterized SAV3 strains, no less than 27 years ago.

## 4. Discussion

PD remains a highly problematic and persistent viral disease in Atlantic salmon aquaculture, diagnosed at 158 different farming locations in Norway during routine PCR surveillance in 2020 [36]. Using genomic epidemiology approaches to study SAV as the causative agent, we have reconstructed scenarios involving multiple independent PD transmission events across different production regions. Our results strongly implicate farms in Trøndelag and Møre og Romsdal as a common source for SAV2a and 2b transmission outside the current zone of SAV2 toleration. This includes independent introductions of SAV2a and 2b into Sognefjord within Vestland, and of SAV2a into Rogaland, in 2019. These events led to the establishment of SAV2 control zones aiming to eradicate the virus outside the toleration zone [36]. In each case, the inferred spread of SAV2 occurred in the opposite direction to the prevailing Norwegian current, and in the case of the proposed transmission of SAV2a to Rogaland (Figure 5A,B), in an inferred short timeframe involving a huge physical distance. Both factors are inconsistent with passive diffusion scenarios and instead imply repeated horizontal spread via anthropogenic routes, likely through well boat activity as a known risk factor for PD transmission [55,56]. In contrast, reconstructed scenarios for Northward spread of SAV2b outside the SAV2 toleration zone—namely the inferred single introduction of SAV2b from Fosen to sites in Northern Trøndelag in 2019 (Figure 6A,B), is consistent with passive diffusion facilitated by ocean currents, but cannot exclude active transmission via a well-boat or other vector.

SAV3 genomes sequenced in this study were a component of SAV2a-SAV3 subtype co-infections from Rogaland (Figure 7) within the SAV3 toleration zone (Figure 1). The short temporal period separating the introduction of SAV2a to Rogaland from Møre og Romsdal in late 2019 (Figure 5A) to the identification of SAV2a-SAV3 co-infections in Rogaland in early 2020 indicates that few barriers exist to co-infections arising. This is consistent with a recent study that documented a PD outbreak involving both SAV2 and SAV3, where around 10% of the infections were SAV2-SAV3 co-infections according to PCR [57]. Another study identified SAV co-infections involving other subtype combinations [14]. It remains unknown if co-infections have significance in terms of pathological outcomes or PD vaccine efficacy, but future studies should consider such issues, while additional efforts may be needed to ensure co-infections are captured in routine PD surveillance. Our discovery of novel strains of SAV3 ancestral to all previously characterized genomes (Figure 7) is notable, as it implies considerable uncharacterized genetic diversity for this subtype, arguing for increased genome sequencing efforts in the SAV3 toleration zone.

Viral whole-genome sequencing has become a routine and cost-effective scientific approach that has gained much prominence during the human COVID-19 pandemic. Considering the extensive commercial importance of salmon aquaculture, balanced against the scale of financial, welfare, and reputational costs caused by viral diseases, we argue that efforts are required to develop and sustain routine viral genomic surveillance programs as part of the ‘toolbox’ to limit viral disease outbreaks [8]. Data such as that reported here, ideally expanded in spatial and temporal resolution, can be used by producers and regulating authorities to support a range of actions aimed at disease control. Moreover, the sequencing approach utilized here and in related studies, e.g., [13,14,16,17], can be readily transferred to other salmonid viruses. Beyond the identification of candidate disease transmission routes, salmonid viruses warrant routine genomic surveillance to capture variants of concern. This is emphatically illustrated by the recent discovery of novel infectious pancreatic necrosis virus (IPNv) strains that have may have evolved to evade long-standing anthropogenic control measures [17,58], including an IPN resistance QTL that has been extremely successful in limiting IPN outbreaks [9].

Finally, the conclusions made in this study were made possible through the cooperation of multiple Atlantic salmon producers sharing PD-infected samples for sequencing. Such a broad ‘buy-in’ from within the aquaculture industry will be essential to achieve and maximize the industry-wide benefits of viral genomic surveillance and epidemiology in the future, ensuring appropriate representation and spatial-temporal resolution of infected samples, essential for powerful reconstructions of viral transmission and evolution in support of disease control actions.

## Figures and Tables

**Figure 1 viruses-13-02549-f001:**
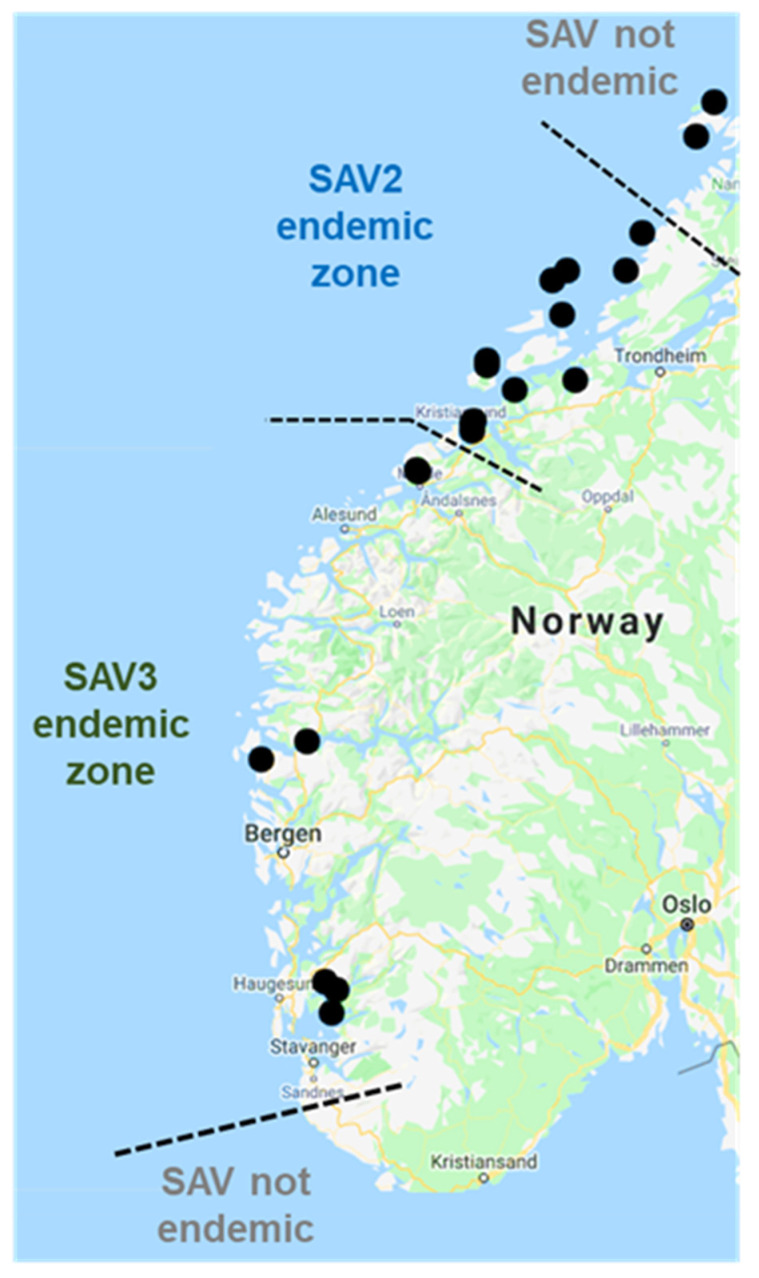
Sampling locations mapped against the current definition of SAV endemic zones.

**Figure 2 viruses-13-02549-f002:**
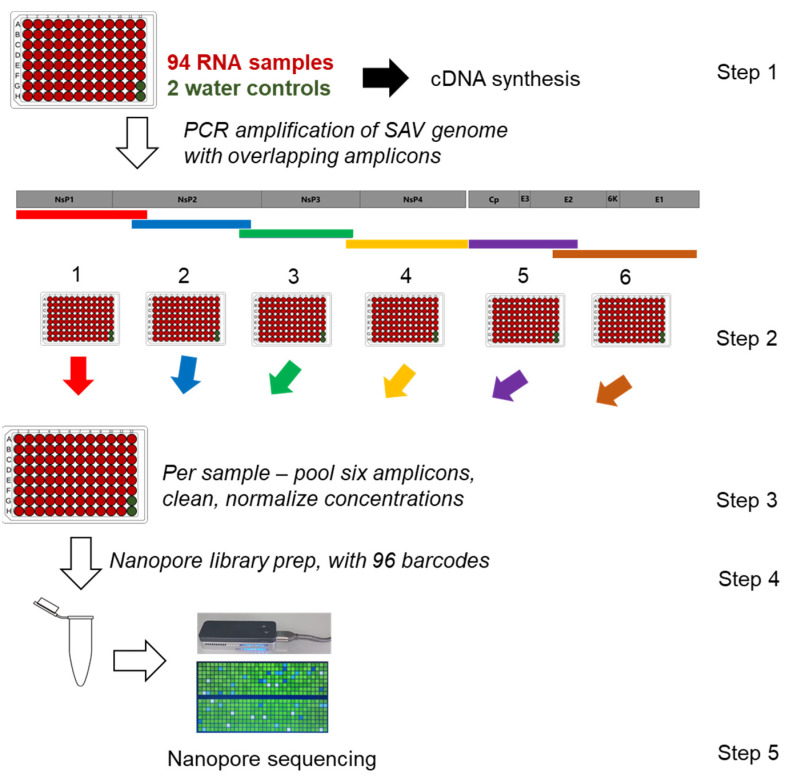
Visualization of steps used in this study generate nanopore sequencing data spanning the SAV genome. See Methods (Section 2.1, Section 2.2 and Section 2.3) for full details.

**Figure 3 viruses-13-02549-f003:**
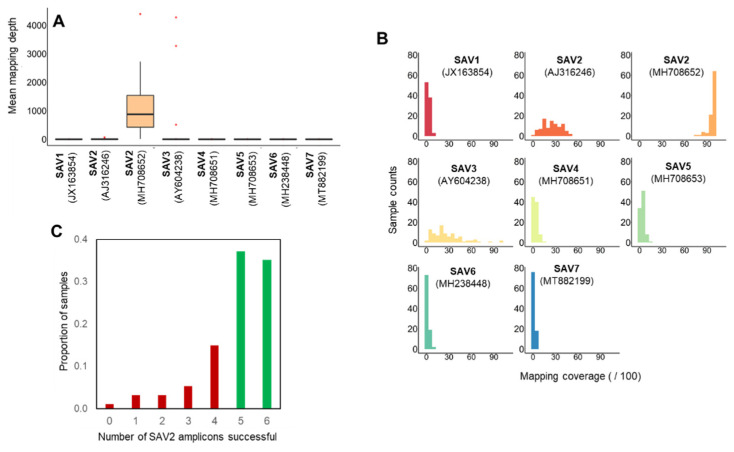
Summary statistics for SAV nanopore sequencing data. (**A**) Boxplot depicting mapping depth across the reference genomes for the seven SAV subtypes (n = 94 samples). Mapping depth represents the number of times the base pairs in the SAV genome are covered by the mapped nanopore reads. (**B**) Frequency histogram depicting the number of samples showing genome-wide coverage against the same reference genomes (bin-widths of 5%). Coverage represents the proportion of base pairs in the reference genome covered by mapped nanopore reads. (**C**). Histogram showing the number of successful amplicons generated (out of 6) as a proportion of n = 94 samples.

**Figure 4 viruses-13-02549-f004:**
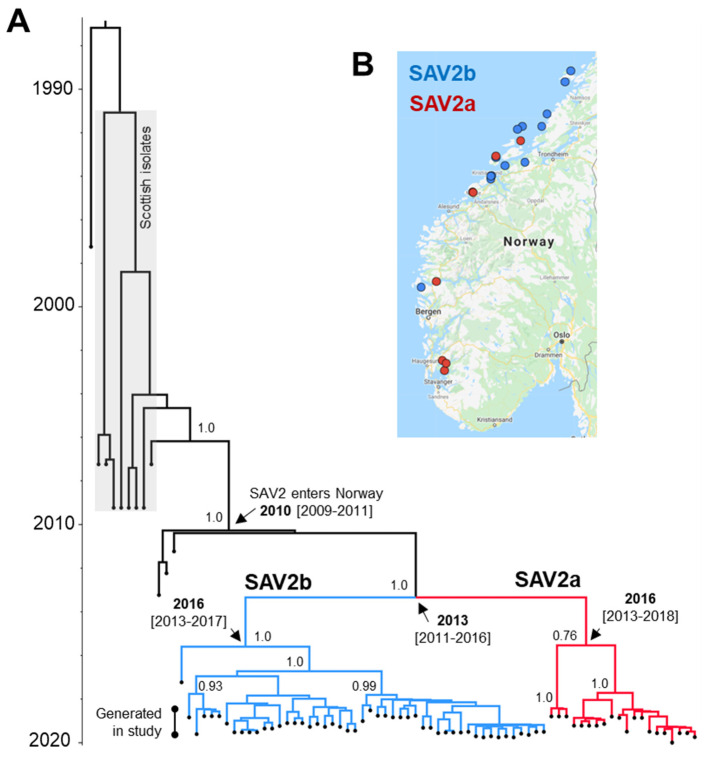
(**A**) Time-calibrated Bayesian phylogeny summarizing the evolution of SAV2 in Norwegian aquaculture. The data represents 81 SAV2 sequences, inclusive of 68 generated by Nanopore sequencing in this study with an average length of 10,600 bp (Appendix A) and 14 additional SAV2 genome sequences (Appendix A). The tree was generated in BEAST2 using a relaxed molecular clock model calibrated with the tip sampling dates, a coalescent prior assuming constant population size and phylogeographic model to reconstruct the most likely geographic location for each node (data from phylogeographic model summarized in Figure 5 and Figure 6). Information is summarized at key nodes, including inferred median node age, with the associated 95% highest posterior density (HPD) around node ages given in parentheses. Posterior probability (PP) values for key nodes are shown (maximum PP = 1.0). A complete representation of the tree, inclusive of 95% HPDs on node ages and with tip sample identifiers, is provided in Appendix A. The sequence alignment is provided in Appendix A. (**B**) Geographical location of SAV2a and 2b sequences inferred by phylogenetics according to the sampling site.

**Figure 7 viruses-13-02549-f007:**
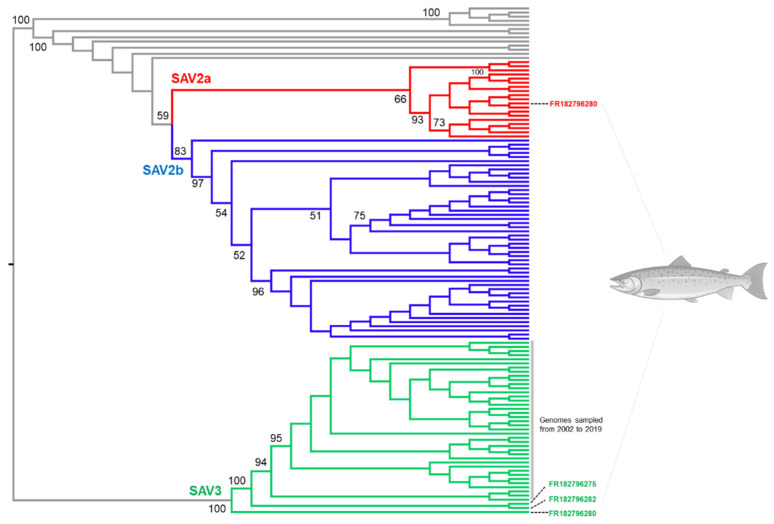
Maximum likelihood phylogeny reconstructing the evolution of SAV2 and SAV3 in Norwegian aquaculture, characterizing the phylogenetic position of the novel SAV3 genomes sequenced in this study (Appendix A). The other included sequences represent 39 SAV3 genomes available on NCBI (Appendix A) and the 81 SAV2 sequences used elsewhere in this study (Appendix A, i.e., Figure 3, Figure 4 and Figure 5). The tree was generated using IQTREE [52] with the best-fitting nucleotide model (TIM2+F+G4). An ultrafast bootstrapping algorithm [53] was used to generate node support values (maximum value: 100). The sequence alignment is provided in the Appendix A.

**Table 1 viruses-13-02549-t001:** Primer sequences used to amplify SAV genomic regions that were nanopore sequenced.

Amplicon	Forward Primer (5’–3’)	Reverse Primer (5’–3’)	Product Length ^1^	Genomic Location ^1^
1	AGACTGCGTTTCCAGGGTTC	CCCGTAGATGCCAATCGTGT	2162 bp	89–2251 bp
2	GAATACGTTTACGAATTGTCCTCC	ACCGAGACGGACTTGAAATACC	1967 bp	2083–4049 bp
3	GACCTGGTGTTTTGTGACGC	TCCCGTGTTAGCCCTCTAGG	1895 bp	3838–5732 bp
4	GCAGCGTCCACRGCCATAGT	CATCAGGCGTTTTACAGGGTC	2015 bp	5531–7545 bp
5	ATGTTTCCCATGCAATTCACCAAC	GGTGCGGCTTGCCCTGGGTGAT	1816 bp	7818–9633 bp
6	AGAGAACGCAGCAAGGGC	GGCACTTCTTCACCACGCA	2431 bp	9318–11,748 bp

^1^ Product length and location defined against the genome sequence of SAV2 isolate SCO07-4619 [37], NCBI accession number: MH708652 (length: 11,780 bp). The relative length and position of each amplicon against the same SAV2 genome is shown in Figure 2.

## Data Availability

Seventy-five novel SAV genomes generated in this study and used in the presented phylogenetic analysis were deposited in the National Center for Biotechnology Information (NCBI) database with accession numbers OL754508–OL754582. The same sequences, along with an additional 25 partial SAV2 genomic sequences generated in the study but not used in phylogenetic analysis, are provided in Appendix A.

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
