# Peer review of "Genomic Epidemiology of Salmonid Alphavirus in Norwegian Aquaculture Reveals Recent Subtype-2 Transmission Dynamics and Novel Subtype-3 Lineages"

_viruses, 2021, doi:10.3390/v13122549_

Round 1

Reviewer 1 Report

I have an opportunity to read the MS entitled “Genomic epidemiology of salmonid alphavirus in Norwegian aquaculture reveals recent subtype-2 transmission dynamics and novel subtype-3 lineages” by Macqueen. This MS showed a large epidemiological study of salmonid alphavirus (SAV) in Norway using nanopore sequencing.

I find the study to be scientifically sound and the data is very nicely presented to show the epidemiology dynamics of SAV transmission. The description is also logically and clearly structured based on experimental data. The authors must be commended for their thorough approach to this study.

I would like to request the author to make only one minor improvement. In introduction section, please add the description about the features of nanopore sequencing technology and its advantages in this research.

Author Response

We thank the reviewer for their positive comments on the quality of our work.

As requested, we have updated the introduction section providing one new sentence on the features of nanopore sequencing technology, and its advantages in this research: “Our previous work has shown that nanopore sequencing of SAV and other fish viruses offers a rapid and cost-effective approach with extremely high consensus accuracy, which can be used to identify complex infections involving different viral strains [37, 14].”

Please note, we also cited articles and reviews in the introduction providing more information about nanopore sequencing and its applications/advantages for this type of research.

Reviewer 2 Report

Salmonid alphavirus (SAV) is a major pathogen of salmon in Northern Europe. This article presents a study of the geographical distribution and the genetic relationships between SAV2 and SAV3 subtypes in several regions in Norway, trying to shed light on the possible origin of the introduction of new subtypes of SAV into certain areas. In the past years, there have been a number of publications dealing with phylogenetic analysis of SAV sequences, but since evolution of viral populations is an ever-changing issue, updates on the situation should be welcome. The paper in here contains some interesting new findings such as the identification of novel strains of SAV3 and SAV2+SAV3 co-infections in salmon. Data seem well supported by nanopore next-generation sequencing, a powerful tool to do genetic analysis of viral populations.

In this paper, the authors work on total RNA samples originally taken from salmon heart tissue from Norwegian aquaculture facilities. On this regard, I was wondering if the salmon fish sampled in this study had been vaccinated vaccinated against SAV. If that was the case: is there any chance of the genetic material in the vaccine showing up after nanopore sequencing?

Also, from the perspective of SAV subtypes dynamics: is the anti-SAV vaccine used in Norway against SAV2, against SAV3, or both?

If I´m not wrong, SAV3 emerged in Norway aquaculture earlier than SAV2. Is there any indication of SAV2 being replacing SAV3 over the years? Are both SAV subtypes considered equal in terms of viral fitness?

In the Discussion section the potential role of wild species reservoirs in the spreading of SAV strains is not discussed.

Author Response

Firstly, thanks for the nice overview of our study and supportive comments. We are happy to address your questions:

"In this paper, the authors work on total RNA samples originally taken from salmon heart tissue from Norwegian aquaculture facilities. On this regard, I was wondering if the salmon fish sampled in this study had been vaccinated against SAV. If that was the case: is there any chance of the genetic material in the vaccine showing up after nanopore sequencing?"

Response: Some of the samples came from vaccinated fish. It is however extremely unlikely that the sequences in the study originate from the vaccine. The vaccine is generally not detectable in the heart, especially this long after vaccination. If, however, the vaccine had been sequenced, it would be easily detected due it being very different in sequence from the contemporary SAV2 and SAV3 sequences in Norway. The three vaccines that are used in aquaculture contain either a SAV1 antigen (a different subtype), a SAV3 antigen (quite different from the SAV3 sequences in the paper, which are novel) or a Scottish SAV2 antigen (also very distinct from Norwegian SAV2s). We feel there is basically zero chance that sequencing of the vaccine has occurred in our study.

"Also, from the perspective of SAV subtypes dynamics: is the anti-SAV vaccine used in Norway against SAV2, against SAV3, or both?"

Response: See above. The commercial vaccines are based on SAV1, SAV2 and SAV3 antigens.

"If I´m not wrong, SAV3 emerged in Norway aquaculture earlier than SAV2. Is there any indication of SAV2 being replacing SAV3 over the years? Are both SAV subtypes considered equal in terms of viral fitness?"

Response: You are correct that SAV3 emerged first. However, SAV3 and SAV2 are endemic in different regions of Norway. So there is no indication that they usually have chance to compete. Co-infections certainly occur when the opportunity arises, as our paper documents, but that is not a common situation currently, as the subtypes normally show quite different geographical distribution. There is a general view that SAV2 may be less virulent, but this has not been confirmed in controlled studies, and could also be an artefact of the difference in geography.

"In the Discussion section the potential role of wild species reservoirs in the spreading of SAV strains is not discussed."

Response: We thank the reviewer for this suggestion, but do not feel its justified to revisit wild species reservoirs in explaining SAV spread in the discussion section. Our results provide no evidence that wild reservoirs were involved in SAV transmission, the data being consistent with anthropogenic spread or passive diffusion from salmon farms. We clearly introduced the potential of wild reservoir species in the introduction, and feel that is sufficient for this article, where the topic is not a major focus due to a lack of data from wild fish.